# Fine Mapping of Candidate Gene Controlling Anthocyanin Biosynthesis for Purple Peel in *Solanum melongena* L.

**DOI:** 10.3390/ijms25105241

**Published:** 2024-05-11

**Authors:** Kai Xiao, Feng Tan, Aidong Zhang, Yaru Zhou, Weimin Zhu, Chonglai Bao, Dingshi Zha, Xuexia Wu

**Affiliations:** 1Shanghai Key Laboratory of Protected Horticultural Technology, Horticultural Research Institute, Shanghai Academy of Agricultural Sciences, Shanghai 201403, China; 15562791500@163.com (K.X.); xiaokai_1993@126.com (F.T.); family20082008@sina.com (A.Z.); yrzhou153@163.com (Y.Z.); wmzhu69@126.com (W.Z.); dingshizha@aliyun.com (D.Z.); 2Institute of Vegetable Research, Zhejiang Academy of Agricultural Sciences, Hangzhou 310021, China

**Keywords:** anthocyanin, eggplant, peel color, candidate gene

## Abstract

Fruit color is an intuitive quality of horticultural crops that can be used as an evaluation criterion for fruit ripening and is an important factor affecting consumers’ purchase choices. In this study, a genetic population from the cross of green peel ‘Qidong’ and purple peel ‘8 guo’ revealed that the purple to green color of eggplant peel is dominant and controlled by a pair of alleles. Bulked segregant analysis (BSA), SNP haplotyping, and fine genetic mapping delimited candidate genes to a 350 kb region of eggplant chromosome 10 flanked by markers KA2381 and CA8828. One ANS gene (*EGP22363*) was predicted to be a candidate gene based on gene annotation and sequence alignment of the 350-kb region. Sequence analysis revealed that a single base mutation of ‘T’ to ‘C’ on the exon green peel, which caused hydrophobicity to become hydrophilic serine, led to a change in the three-level spatial structure. Additionally, *EGP22363* was more highly expressed in purple peels than in green peels. Collectively, *EGP22363* is a strong candidate gene for anthocyanin biosynthesis in purple eggplant peels. These results provide important information for molecular marker-assisted selection in eggplants, and a basis for analyzing the regulatory pathways responsible for anthocyanin biosynthesis in eggplants.

## 1. Introduction

Eggplant (*S. melongena* L.) is a vegetable that is distributed worldwide and is rich in vitamins, has a good taste, and is edible [1]. The color of eggplant is not only used as an evaluation standard for fruit ripeness but also as an important factor affecting consumers’ purchase choices. Eggplant fruit color is mainly attributed to chlorophyll and anthocyanins. Anthocyanins are water-soluble secondary metabolites derived from the flavonoid biosynthesis pathway, and are thought to be antioxidant compounds that play an important roles in reducing cardiovascular diseases and preventing certain types of cancer and degenerative diseases in humans [2,3]. Anthocyanins can present a wide range of colors; therefore, they play a very important role in the formation of fruit color, and are related to adaptability to light, temperature, and other environmental conditions [4]. In eggplants, anthocyanins accumulate for a long time, lasting until physiological fruit ripening, when they begin to degrade and fade. Low light or high temperatures in greenhouses often lead to color differences in eggplant, thus reducing the apparent quality of eggplant [5]. Therefore, a comprehensive understanding of the key genes involved in anthocyanin synthesis in eggplant fruits will provide important guidance for studying anthocyanin accumulation.

At the molecular level, anthocyanins are biosynthesized through the phenylpropane (phenylalanine) pathways [3,6] which contain the following key enzyme genes: CHS, encoding chalcone synthase; CHI, encoding chalcone isomerase; F3H, encoding flavanone 3β-hydroxylase; F3′H, encoding flavonoid 3′-hydroxylase; F3′5′H, encoding flavonoid 3′,5′-hydroxylase; DFR, encoding dihydroflavonol 4-reductase; and ANS, encoding anthocyanidin synthase [7]. Phenylalanine is catalyzed by dihydroflavonols and then transformed into anthocyanidins and proanthocyanidins through colorless leucoanthocyanidins [8]. Through the actions of CHS and CHI, three malonyl-CoA molecules and one *p*-coumaroyl-CoA molecule condense to produce naringenin [9,10,11]. Naringin is converted to dihydroxanthol under the action of F3H and then catalyzed by F3′H and F3′5′H to form dihydromyricetin and dihydroquercetin [12,13]. DFR catalyzes the reduction of dihydromyricetin and dihydroquercetin to colorless anthocyanins, which are subsequently transformed into colored anthocyanins by ANS eventually [14,15,16].

The expression levels of *SmCHS*, *SmCHI*, *SmF3H*, *SmF3′5′H*, *SmDFR*, and *SmANS* were positively correlated with anthocyanin content, which was higher in purple eggplants than that in white eggplants [17]. Structural genes for anthocyanin pathway enzymes are regulated by the MBW complex, composed of R2R3-MYB, bHLH, and WD40 regulators [7]. To date, large quantities of R2R3-MYB proteins regulating anthocyanin biosynthesis [18] have been identified in Arabidopsis [19], apples [20], tomatoes [21], grapes [22], potatoes [23], and eggplants [24]. In eggplant, *SmMYB1* [17], *SmMYB35* [25,26], *SmMYB*113 [27], and *SmMYB* 75 [28] have been positively correlated with anthocyanin synthesis in fruits, whereas *SmMYB44* and *SmMYB86* have been identified as negative regulators [25]. The WD and bHLH transcription factors have been studied less frequently studied than MYB in eggplants. *SmbHLH13* may promote anthocyanin accumulation through the positive regulation of *SmCHS* and *SmF3H* [25]. *SmbHLH1* was identified as a negative regulator of anthocyanin biosynthesis and directly represses the expression of *SmDFR* and *SmANS* expression [29]. The WD-type gene *SmTTG1* interacts with the bHLH-type gene *SmTT8* and both *SmTTG1* and *SmTT8* interact with *SmMYB* to form a complex regulating anthocyanin accumulation [30]. And the photochrome gene HY5 could promotes anthocyanin synthesis by binding *to SmCHS* and *SmDFR* [31].

Mutations in the coding or promoter regions of the anthocyanin gene are associated with changes in fruit color. Differences in *SmMYB1* promoters between photosensitive eggplants [17] and non-photosensitive eggplants [24] may lead to differences in the light dependence of anthocyanin synthesis. In apples, a single nucleotide polymorphism (SNP) in the MYBA promoter region results in abnormal anthocyanin synthesis, and the expression level of anthocyanins in dark red varieties is significantly higher than that in light red varieties [32]. The sequence variety of the *Capana10g001978* promoter in peppers may be responsible for the purple mature unripe fruit and green mature unripe fruit [33]. Doganlar et al. [34] and Frary et al. [35] demonstrated the existence of locus *fap10.1* and *fap10.2* on chromosome 10 with or without close linkage to anthocyanin, and the contribution rate of *fap10.2* was as high as 100%. However, to date, no co-segregated markers or candidate genes for explaining the presence or absence of anthocyanins in eggplant have been reported so far.

In this study, we investigated the inheritance of eggplant peel color in F1, F2, and BC1 populations derived from a cross of ‘Qidong’ (green) and ‘8 guo’ (purple) lines. Molecular markers have been developed to screen for recombinants and narrow down the initial region to achieve fine localization. An SNP in *SmANS* (*EGP22363*), encoding a 2-ketoglutarate dependent dioxygenase, was shown to be responsible for the green color in ‘Qidong’. Cloning and functional analysis of *SmANS* is helpful for understand the formation mechanism of green peel in eggplant and to promote the breeding of green peel eggplant varieties.

## 2. Results

### 2.1. Pigment Content and Genetic Analysis of Color Traits of Eggplant Peel

The green peel eggplant ‘Qidong’ was used as the maternal parent P1, and the purple eggplant ‘8 guo’ was used as the paternal parent P2 to obtain the F1 generation population, whose peel color was purple (Figure 1a). The analysis of main pigment content showed that the content of anthocyanins in ‘Qidong’ remained at a low level in all periods. The content of anthocyanins in ‘8 guo’ showed a trend of first increasing and then decreasing, reaching a peak value at day 25, and the content in each period showed significant differences compared with that in ‘Qidong’ (Figure 1a,b). The chlorophyll content of ‘Qidong’ first increased and then decreased, and maintained a high level from 15 d to 30 d (Figure 1c). The chlorophyll and carotenoid contents of ‘8 fruit’ increased gradually with time, and reached the peak on the 35th day. The chlorophyll content of ‘8 fruits’ at 15 d to 25 d was significantly lower than that of ‘Qidong’ at the same period (Figure 1b,c). Due to the presence of some intermediate colors of 404 F2 progenies that were not consistent with the parent, when classifying for color, plants that were completely devoid of purple in fruit peel were recorded ‘green’, otherwise ‘purple’. The fruit color analysis of 404 F2 progenies showed that 300 and 104 plants had purple and green peels, respectively, with a purple/green ratio = 2.88 (χ^2^ = 0.059, *p* > 0.05), which was consistent with the Mendelian inheritance theory of a 3:1 ratio of single-gene controls (Table 1). The BC1P1 population was obtained by backcrossing F1 with the parents of the green eggplant line, including 20 purple eggplant plants and 16 green eggplant plants, with a purple/green ratio = 1.25 (χ^2^ = 0.223, *p* > 0.05), which is consistent with the 1:1 ratio of single-gene controls in Mendelian inheritance. The BC1P2 generation population was obtained by backcrossing the F1 with the purple-line parent, and the peel color was purple (Table 1). From the above genetic manifestations, it can be seen that purple and green peels were dominant and controlled by a pair of alleles in the eggplant peel in this study population.

### 2.2. Preliminary Mapping of Candidate Gene via Bulked Segregant Analysis (BSA)

To determine the candidate gene intervals associated with the purple peel trait in eggplant, 25 plants each from purple- and green-colored fruit were selected from the F2 population to construct a mixed pool. The Illumina NovaSeq sequencing platform was used to re-sequence the parents and the mixed pool. The sequencing depth was 19.71×, 19.45×, 38.65×, and 38.85×, respectively, and the coverage reached more than 98% (Appendix A). The clean reads from each sample were aligned to the eggplant reference genome at an alignment rate of >99%. A total of 1,762,409 SNP sites were identified. Four algorithms, ED4 (Euclidean distance raised to the fourth power), G-value (a smoothed version of the standard G statistic), LOD, and SNP ratio, were used to locate the candidate interval in the 0.88 Mb region (2.91 Mb–3.79 Mb) on chromosome 4 (Figure 2) and the 0.8 Mb region (6.61 Mb–7.41 Mb) and 1.2 Mb region (19.11 Mb–20.31 Mb) on chromosome 10 (Figure 2), with 199 candidate genes in the candidate interval.

### 2.3. Fine Mapping of the Candidate Gene

Based on the results of BSA-seq, 92 pairs of KASP, Indel, and dCAPs markers were designed and developed using the information on differential loci between parents and offspring in the candidate interval, including the indel and SNP loci. KASP genotyping results showed that the markers KA2381 and KA2388 performed well, and the candidate interval was determined to be the 1.2 Mb region (19.11 Mb–20.31 Mb) on chromosome 10 (Figure 3). Polymorphic markers in the parents and F1 were subsequently validated in the F2 population of 359 individual plants, of which 11 showed stable polymorphisms in the F2 population. The Indel markers and cleaved amplified polymorphic sequence (CAPS) markers were designed for this candidate region. The total length of the linkage map on chromosome 10 was 4.6 cm, the number of markers was 11, and the average distance between the marker was 0.42 cm. The candidate gene was located at a position of 0.1 cm between KASP marker KA2381 and dCAPs marker CA8828 (Figure 4). The physical distance was reduced to an interval of 0.35 Mb (19.20 Mb–19.55 Mb).

### 2.4. Sequence and Expression Analysis of the ANS Candidate Gene

The results showed that there were 656 SNPs and 110 indels in the candidate interval, but most of these SNPs and indels were located in the spacer and intron regions of genes. Only six SNPs and one indel were located in the exon region of genes, which included an SNP in the EGP22363 gene region (Appendix A). There were 16 genes in the candidate intervals on chromosome 10 (Table 2). EGP22363 encodes 2-oxoglutarate-dependent dioxygenase (2-OGD). Sequence alignment using NCBI database revealed that it is highly homologous to ANS, a member of the 2-OGD superfamily in tomatoes (XP_010312366), which plays a key role in the flavonoid metabolic pathway. The DNA and promoter sequences of the candidate gene, EGP22363, were obtained from the eggplant reference genome. The DNA sequence was 1469 bp in length, and contained three exons and two introns (Figure 5a). After alignment with the reference genome, we found that the DNA sequence of the purple parents was completely consistent with that of the reference genome, whereas the green parents had an SNP at 1435 bp in the third exon (Figure 5b). The promoter sequences of green and purple parents were consistent.

An amino acid sequence analysis of the candidate gene EGP22363 showed that a T-C mutation at position 1435 bp changed hydrophobic leucine (TTA) to hydrophilic serine (TCA) (Appendix A). The secondary and tertiary structures of the candidate gene proteins were predicted, and the gene mutation was found to increase the coil structure in the secondary structure of the protein and change its tertiary spatial structure (Figure 5c). To reveal the expression of EGP22363 gene in eggplant peel, the eggplant peel samples of ‘Qidong’ and ‘guo’ at the early stage of growth and development, i.e., 16 and 24 days after pollination (DAP), were used for quantitative PCR (Figure 6). For ‘Qidong’ and ‘8 guo’, the expression level of ‘8 guo’ was significantly higher than that of ‘Qidong’ in the early stage of growth and development. Therefore, SmANS might regulate anthocyanin biosynthesis. We hypothesized that the EGP22363 mutation affected the function of the original protein, thereby restricting the synthesis of anthocyanins and leading to the green color of eggplant.

### 2.5. Development of dCAPS Marker

The promoter sequences of green and purple parents were consistent. Based on the differential loci of the candidate genes in the two parents, a dCAPs molecular marker was developed to distinguish between purple- and green-peeled eggplants (Figure 7a). The molecular markers were screened in parental materials ‘Qidong’, ‘8 guo’, and the F1 generation, and the effectiveness of the markers was verified in 42 inbred lines and 24 hybrid varieties (48 purple and 18 green eggplants, Appendix A). As shown in Figure 3b, the verified accuracy rate was 98.5% for the breeding lines, with one green line exhibiting mismatched genotypes and phenotypes (Figure 7b,c).

## 3. Discussion

The peel color of eggplant is one of the most intuitive horticultural traits, being mainly determined by anthocyanins and chlorophyll and affected by various environmental factors. The purple color of eggplant fruit can be divided into purple red, purple black, and other different degrees, the quantitative traits of which are controlled by multiple genes. In this study, we used eggplant ‘Qidong’ with green skin and eggplant ‘8 guo’ with purple skin (Figure 1b) to conduct a genetic analysis of eggplant fruit color. The color of the fruits in the F2 population was not uniform, and can be subdivided into purple red, purple black, light green, etc., but in the genetic analysis, we classified them as purple or green. Based on the segregation of the offspring population, it was inferred that the purple and green fruit colors of eggplants were dominant and controlled by a pair of alleles (Table 1).

The localization results of BSA-seq showed that three peaks on chromosomes 4 and 10 exceeded the threshold, and the candidate interval was identified as the 1.2 Mb region on chromosome 10 based on the KASP genotyping results (Figure 2 and Figure 3). Subsequently, the locus were mapped to a 350 kb interval in which there were 16 genes (Figure 4). After annotating the candidate genes, we speculated that *EGP22363* is the most likely functional gene encoding 2-ketoglutarate dependent dioxygenase (2-OGD) (Table 2). The 2-OGD superfamily is one of the largest protein families in plants, and it plays an important role in the flavonoid pathway [36]. A sequence analysis of *EGP22363* in the NCBI database showed that it is highly homologous to ANS, a member of the 2-OGD superfamily in tomatoes (XP_010312366.1). ANS is a key enzyme that directs the flavonoid pathway into a branch of the anthocyanin synthesis pathway [37] and plays a crucial role in the production of anthocyanins. The transcript levels of genes related to anthocyanin synthesis in purple and black eggplants were significantly higher than those in white eggplants, and the difference in ANS transcript levels of ANS is the greatest, indicating that ANS may play a key role in the formation of purple eggplants [38]. Overexpression [39,40] and silencing [41] of ANS genes have also been shown to affect the final formation of colored anthocyanins, thereby altering fruit color.

The protein structures of the candidate genes were predicted, and the key features of the DIOX_N domain and IPNS-like were found in anthocyanin synthesis-related gene proteins of Hibiscus [42] and other plants, indicating that *EGP22363* may play a role in anthocyanin biosynthesis. An analysis of the *EGP22363* sequence in the purple and green parents revealed an SNP at position 1435 in the third exon, located in the third exon of the *EGP22363* DNA sequence. The T-C base variation causes the leucine (TTA) of the purple parent amino acid sequence to change to serine (TCA) in the green parent during translation, and the secondary and tertiary structures of the protein are also changed (Figure 5a–c). The structure of a protein determines its function; therefore, this base variation may have affected the function of *EGP22363*, resulting in loss of anthocyanins.

The relative expression level of the candidate gene, *EGP22363*, was significantly higher in purple than in green eggplants, indicating that this gene may play an important role in eggplant fruit color (Figure 7a). The expression levels of genes are affected by various factors such as promoters and transcription factors. For example, the insertion and deletion of cis-acting elements in the promoter region of an anthocyanin synthesis gene in pepper resulted in a significant difference in its expression between purple and green fruits [33]. However, in this study, we found no difference in the promoter sequence of *EGP22363* between the two parents, whereas the relative expression levels of anthocyanin-related transcription factors were significantly different, indicating that the expression of *EGP22363* in purple and green eggplants may be regulated by transcription factors (Appendix A). Therefore, the protein structure and expression level of this gene may jointly affect the synthesis of anthocyanins, resulting in the purple or green appearance of eggplant. However, the mutual relationship between the two needs to be further studied.

## 4. Materials and Methods

### 4.1. Plant Material, Phenotyping, and Segregation Analysis

The female parent Qidong (producing green peel fruits) and male parent ‘8 guo’ (producing purple peel fruits) were used to map the genes regulating the purple character of eggplant fruits. ‘Qidong’ and ‘8 guo’, high-generation inbred lines developed in our lab, produced short rod-like shaped fruit. The F1 (producing purple fruit-producing) plants were self-crossbred to obtain 404 F2 plants. Color phenotypic data were obtained by visual observation during the growth and development stages of the fruit. The plants with a complete absence of purple color of fruit peel in F2 progenies are recorded as ‘green’, otherwise ‘purple’. Plants were planted in a Zhuanghang Experimental Station greenhouse at the Shanghai Academy of Agricultural Sciences (Shanghai, China). A chi-square test was conducted to verify the separation of the F2 generations using Microsoft Excel.

### 4.2. Determination of Anthocyanin, Chlorophyll, and Carotenoid Content

#### 4.2.1. Anthocyanin

The extraction method was referred to Jeong [43]. 0.25 g of peel tissue was ground with liquid nitrogen and 5 mL of 1% (volume fraction) hydrochloric acid-methanol solution (pre-cooling at 4 °C) was added. Then, the supernatant was extracted at 4 °C for 12 h without light, and the supernatant was obtained by centrifugal filtration. The light absorption values at 600 nm and 530 nm were measured with 1% hydrochloric acid-methanol solution as blank control. The anthocyanin content was expressed by the difference between the absorption values at 530 nm and 600 nm.

#### 4.2.2. Chlorophyll and Carotenoid Content

An amount of 0.25 g of peel was ground with liquid nitrogen and 5 mL of 1% hydrochloric-methanol solution was added. Then, the supernatant was centrifugally taken and the absorbance values were measured at 663 nm, 6r45 nm, and 470 nm with 80% acetone as a blank control. The contents of chlorophyll a, chlorophyll b, and total chlorophyll were calculated according to Porra’s [44] method.

The calculation formula is as follows:Chlorophyll (total content mg/g) = (8.02 × OD663 + 20.21 × OD645) × V/1000 × W.
Chlorophyll a (Ca) = (12.7 × OD663 − 2.69 × OD645) × V/1000 × W.
Chlorophyll b (Cb) = (22.9 × OD645 − 4.68 × OD663) × V/1000 × W.
Carotenoid (mg/g) = (4.367 × OD470 − 0.014 × Ca − 0.454 × Cb) × V/1000 × W.
where V is the volume of the extract; W is the fresh weight of the leaves; Ca is the concentration of chlorophyll a; and Cb is the concentration of chlorophyll b.

### 4.3. DNA Extraction and Bulked Segregant Analysis

DNA was extracted from fresh leaves using the cetyl trimethylammonium bromide (CTAB) method [45]. The DNA quality was determined using a Nanodrop 2000 spectrophotometer (Thermo Scientific, Waltham, MA, USA) and 1% agarose gel electrophoresis. To prepare the bulk DNA, two DNA pools were constructed by mixing equal amounts of DNA from 25 purple peel F2 individuals (purple pool) and 25 green peel F2 individuals (green pool). The Illumina HiSeq™ PE150 platform (San Diego, CA, USA) was used for double-ended sequencing of a mixed pool with 150 bp × 2 paired-end sequences. After sequencing, raw reads were preprocessed to remove adaptors and low-quality reads (bp < 20). Duplicated reads were mapped onto the reference genome of eggplant [46] using BWA software (https://sourceforge.net/projects/bio-bwa/ (accessed on 8 May 2024)) with SAM tools [47]. Single-nucleotide polymorphisms (SNP) and insertion/deletions (indels) were called and filtered by removing heterozygous and missing SNPs as well as indels in the pools and parental lines using the GATK software (https://software.broadinstitute.org/gatk/2 (accessed on 8 May 2024)) [48]. The results of the SNP ratio, ED4, G-value, and smooth LOD [49] algorithms were summarized according to the SNP site information, and the candidate regions for the association of purple peel traits were preliminarily located.

### 4.4. Fine Mapping of the Candidate Gene

Based on the preliminarily located region, KASP markers were designed for each 1–2 Mb distance in the interval from BSA. The PCR reaction of KASP was performed using a 3 μL system, 2 × KASP Master Mix 1.4792 μL, ddH_2_O 1.4983 μL, forward primers 1 and 2 0.005 μL, reverse primers 0.0125 μL, and dry DNA 20–50 ng. The reaction procedure was set as follows: 94 °C, 15 min, 1 cycle; 95 °C, 20 s, 65–56 °C, 60 s, 10 cycles, annealing extension temperature decreased by 0.8 °C for each cycle; 94 °C, 20 s, 57 °C, 60 s, 26 cycles. The fluorescence signals of the PCR products were recorded. We designed nine pairs of KASP markers to analyze 359 plants to identify recombinant plants (Appendix A). Indel and dCAPS markers between the parents were developed to further locate the candidate genes. SNPs were converted to dCAPS markers using ‘dCAPS finder 2.0’ (http://helix.wustl.edu/dcaps/dcaps.html (accessed on 8 May 2024)). The PCR products were electrophoresed on an 8% polyacrylamide gel. Screening of the F2:3 population revealed that the locus is located between KA2381 and CA8828. Based on the phenotypic and marker data, the JoinMap4 software (https://www.kyazma.nl/index.php/JoinMap/ (accessed on 8 May 2024)) was used to create the genetic map.

### 4.5. Cloning and Sequence Analysis of Candidate Gene

Primers for the candidate genes were designed according to the eggplant reference genome using Primer Premier 5.0. Total RNA from fruit peels (16 and 24 days after anthesis) was extracted using a Plant RNA Purification Kit (Accurate Biotechnology, Changsha, China) following the manufacturer’s instructions. First-strand complementary DNA (cDNA) was synthesized using the Evo M-MLV Reverse Transcription Kit (Accurate Biotechnology, Changsha, China). Real-time PCR was performed as follows: 95 °C for 5 min, followed by 40 cycles of 95 °C for 10 s, 60 °C for 10 s, and 72 °C for 10 s. Relative expression values were calculated using the 2^–ΔΔCT^ method [50]. *SmAPRT* was used as an internal control. Three biological replicates were used for each gene and at least three technical replicates were used for each sample. The 2000 bp sequence of *EGP22363* upstream of ATG was cloned from Qidong (producing green fruit) and 8 guo (producing purple fruit). PCR amplification of the promoter and coding sequences were performed using A8 FastHiFi PCR Master Mix (Aidlab, Hongkong, China) and according to the manufacturer’s instructions. Appropriate PCR clones were selected and sent to the Shanghai Shengong Biotechnology Co., Ltd. (Shanghai, China) for sanger sequencing. Sequencing chromatograms were validated by multiple alignments using DNAMAN v.9 software (Lynnon Biosoft, San Ramon, CA, USA). The primers used for these analyses are listed in the Appendix A. The candidate gene protein structure domains were predicted using InterPro (http://www.ebi.ac.uk/interpro (accessed on 8 May 2024)) online.

## 5. Conclusions

The candidate gene regulating purple peel color was located in a 350 kb region on chromosome 10 in *Solanum melongena* L. An ANS gene (*EGP22363*) was predicted to be a candidate gene based on gene annotation, and *EGP22363* was highly expressed in purple peels, indicating that it is a strong candidate gene for anthocyanin biosynthesis.

## Figures and Tables

**Figure 1 ijms-25-05241-f001:**
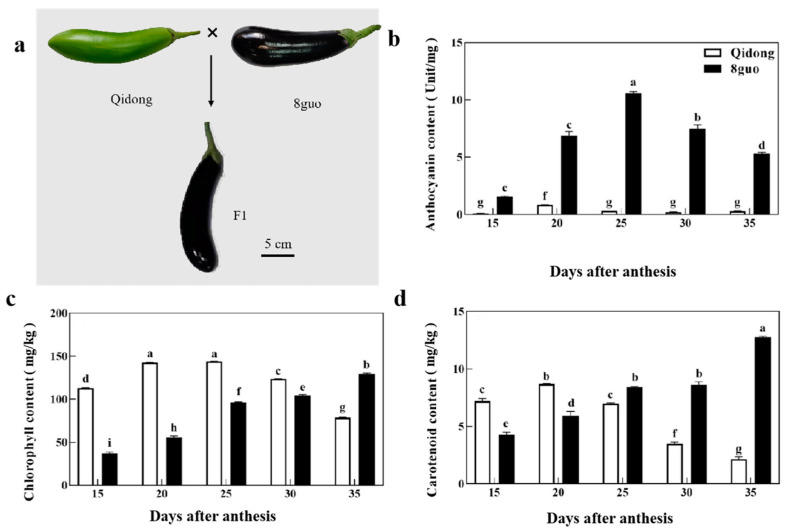
Phenotypic characteristics, pigment content of the parental peel. (**a**): ‘Qi dong’, ‘8 guo’, and F1 fruits. (**b**–**d**): Changes in anthocyanin (**b**), chlorophyll (**c**), and carotenoid (**d**) content in peel during fruit development. Significant differences were determined using ANOVA followed by Tukey’s HSD test (*p* < 0.05). The error bar indicates the standard deviation. Different letters above the bars indicate the statistically significant difference at *p* < 0.05.

**Figure 2 ijms-25-05241-f002:**
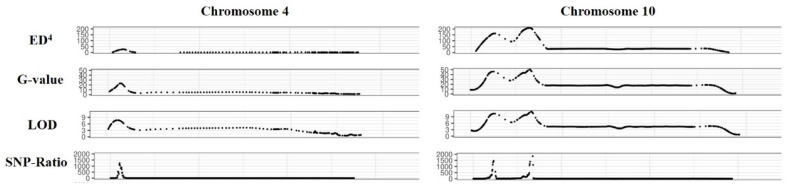
The results of bulked segregant analysis based on four algorithms. The peak is where the target gene is located.

**Figure 3 ijms-25-05241-f003:**
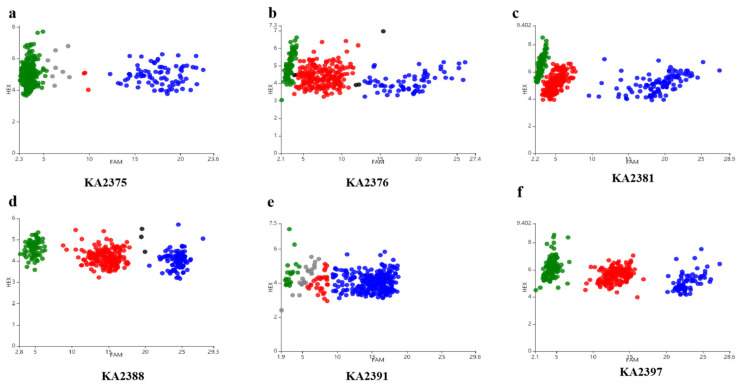
KASP genotyping variations in the F2 population. (**a**,**b**) KASP markers KA2375 and KA2376 in the candidate interval of chromosome 4. (**c**,**d**) KASP markers KA2381 and KA2388 in the candidate interval of chromosome 10 (19.11 Mb–20.31 Mb). (**e**,**f**) KASP markers KA2391 and KA2397 in the candidate interval of chromosome 10 (6.61 Mb–7.41 Mb). Green dot represents pure recessive genotypes, blue represents pure dominant genotypes, red represents heterozygous genotypes, and gray represents no template control (NTC).

**Figure 4 ijms-25-05241-f004:**
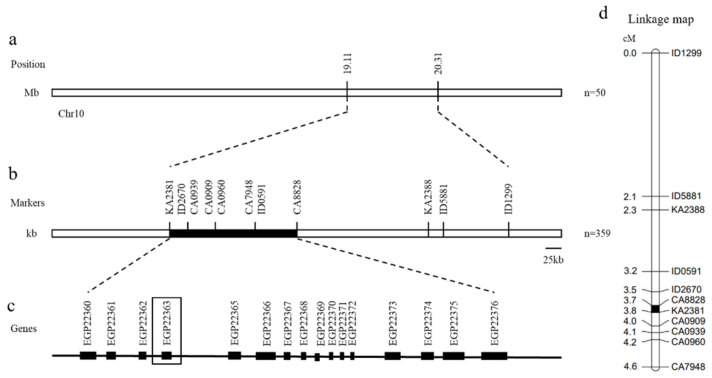
Fine mapping of candidate genes in eggplant. (**a**) The candidate interval obtained by BSA-seq using 50 individual plants contrasting in F2 population. (**b**) The physical map of candidate genes. Positions of the markers are indicated in kb. The candidate genes were located between marker KA2381 and marker CA8828 by using recombinants in 359 individual plants of F2 population. (**c**) Sixteen genes in the candidate interval for fine localization. (**d**) Genetic map of the candidate gene locus. Positions of the markers are indicated in cm.

**Figure 5 ijms-25-05241-f005:**
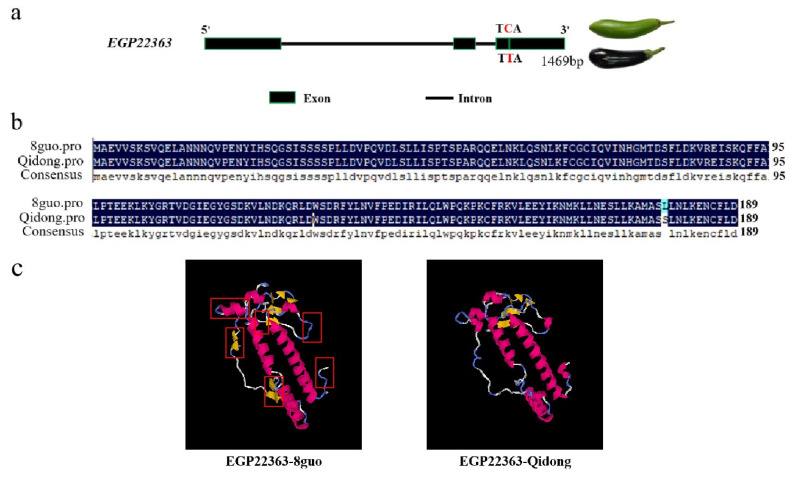
Sequence and structure analysis of the candidate gene. (**a**) Gene structure diagram and nucleotide sequence variation of *EGP22363*. Black rectangles and solid lines represent exons and introns, respectively. The T-C mutation is marked using a red letter. (**b**) The DNA sequence alignment of *EGP22363* between ‘8 guo’ and ‘Qi dong’. (**c**) Tertiary structure prediction of the candidate gene. The red boxes mark the different structural areas.

**Figure 6 ijms-25-05241-f006:**
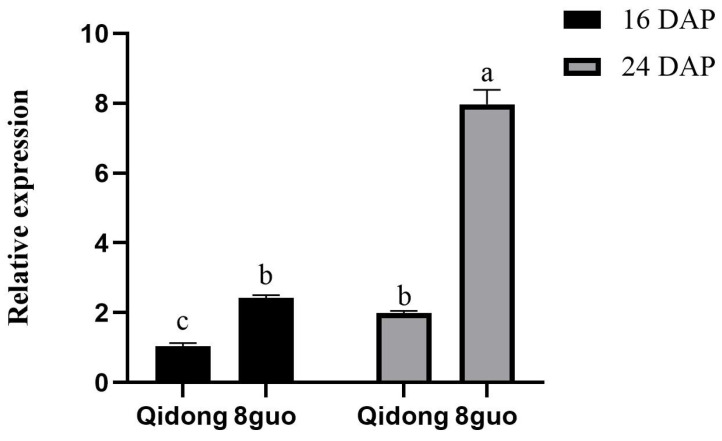
Relative expression of *EGP22363* in female parent ‘Qidong’ and male parent ‘8 guo’ at two development stages. Significant differences were determined using ANOVA followed by Tukey’s HSD test (*p* < 0.05). The error bar indicates the standard deviation. Different letters above the bars indicate the statistically significant difference at *p* < 0.05.

**Figure 7 ijms-25-05241-f007:**
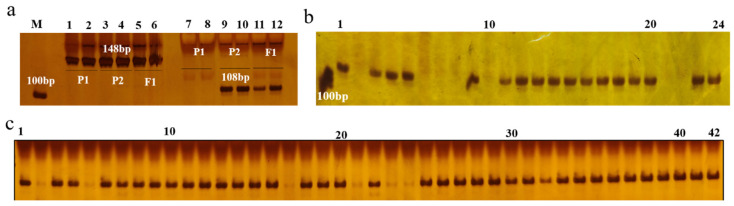
PCR fragments amplified using markers CAPS-7428. (**a**) The amplification results of CAPS-7428 marker in P1, P2, and F1. Lane 1–6: Undigested products. Lane 7–12: digested products. (**b**,**c**) The digested products of 24 hybrid strain lines (**b**) and 42 high-generation inbred lines (**c**).

**Table 1 ijms-25-05241-t001:** Genetic analysis of peel color of F1, F2, and BC1 progenies.

Population	No. of Purple	No. of Green	Total No.	Expected Distribution	χ^2^	*p* Value
F1 ^a^	40	0	40	–		
F2 ^b^	300	104	404	3:1	0.059	0.730
BC1P1 ^c^	20	16	36	1:1	0.223	0.505
BC1P2 ^d^	60	0	60	-		

^a^ F1 = Qidong × 8 guo; ^b^ F2 population was derived from the self-pollination of F1 (Qidong × 8 guo; ^c^ BC1P1 = F1 (Qidong × 8 guo) × Qidong; ^d^ BC1P2 = F1 (Qidong × 8 guo) × 117 8 guo.

**Table 2 ijms-25-05241-t002:** Gene ID, location, and annotation of candidate regions on chromosome 10.

Gene ID	Start (bp)	End (bp)	Annotation
*EGP22361*	19,233,453	19,234,495	Transcription factor DIVARICATA
*EGP22362*	19,243,931	19,247,107	Laccase-4
*EGP22363*	19,251,194	19,258,899	Probable 2-oxoglutarate-dependent dioxygenase (ANS)
*EGP22365*	19,281,490	19,288,169	Codeine O-demethylase
*EGP22366*	19,294,651	19,303,939	Ataxin-3 homolog
*EGP22367*	19,308,677	19,323,779	50S ribosomal protein L9
*EGP22368*	19,328,224	19,330,298	Tubulin beta-1 chain
*EGP22369*	19,334,156	19,337,557	Nr--PREDICTED: uncharacterized protein
*EGP22370*	19,337,569	19,339,613	Fructose-1,6-bisphosphatase
*EGP22371*	19,340,168	19,342,872	Fructose-1,6-bisphosphatase
*EGP22372*	19,346,466	19,347,903	Nr--PREDICTED: uncharacterized protein
*EGP22373*	19,365,049	19,368,599	Nr--PREDICTED: uncharacterized protein
*EGP22374*	19,386,652	19,390,789	Phosphatidylinositol:ceramide inositolphosphotransferase 1
*EGP22375*	19,392,998	19,399,256	F-box protein SKIP8
*EGP22376*	19,400,884	19,401,911	18.2 kDa class I heat shock protein

## Data Availability

The data presented in this study are available in this article and as Appendix A.

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
