# Peer review of "Fine Mapping of Candidate Gene Controlling Anthocyanin Biosynthesis for Purple Peel in *Solanum melongena* L."

_ijms, 2024, doi:10.3390/ijms25105241_

Round 1
Reviewer 1 Report
Comments and Suggestions for Authors
The aims of the paper are certainly of interest for International Journal of Molecular Sciences. The results are interesting and encouraging. However, the results section needs to be reorganized to make it coherent and the discussion has to be improved.
Specific comments follow:
1. Line 69: "Sm" in SmCHS and SmF3H have to be in italics.
2. In line 105 change Fig.1, etc. to Figure 1, etc.
3. The footer of table 1 is different from what appears in the table.
Please replace "b BC1P1 = F1 (Qidong × 8guo) × Qidong" with b F2 population was derived from the self-pollination of F1 (Qidong × 8guo).
Replace "c BC1P2 = F1 (Qidong × 8guo) × 117 8guo" with c BC1P1 = F1 (Qidong × 8guo) × Qidong.
Replace "d F2 population was derived from the self-pollination of F1 (Qidong × 8guo) " with d BC1P2 = F1 (Qidong × 8guo) × 117 8guo
4. The description of the results of Figure 1b-d is missing in the results section. Additionally, the authors did not describe how they measured the content of anthocyanin, chlorophyll, and carotenoid in the Materials and Methods..
5. Please check the font size of the text
6. Line 145 and 146: "F23 population”?? I think authors mean "F2 population".
7. The results section needs to be reorganized, and all Figures and Tables should be inserted into the main text close to their first citation. They must be numbered according to their order of appearance.
8. Supplementary Tables and Figures should be numbred in order in the main text and in supplementary file.
9. The discussion has to be improved
Comments on the Quality of English LanguageTo ensure the utmost quality of the manuscript, some minor revisions in terms of language and grammar are necessary.
Author Response
“Please see the attachment

Reviewer 2 Report
Comments and Suggestions for Authors
This study deals with the fine mapping of the SmeANS gene, which controls anthocyanin biosynthesis for purple skin in unripe eggplant fruit. The inheritance of eggplant color was studied in F1, F2, and BC1 populations derived from two eggplant lines with different skin colors of the unripe fruit. An SNP in SmANS, EGP22363, was found to be responsible for the green color of unripe eggplant fruit.
The paper addresses an important topic; however, a thorough revision of almost all parts of the MS is needed. I have made my comments and edits directly in the MS file, the most important ones are:
· Please state clearly the objectives of your study, mention in an orderly manner what has been done, and what were the main findings.
· I think there are some unnecessary details in the introduction that need to be reduced to focus more on the background and rationale of the study.
· There are Many mistakes and mismatches in the results.
· Some figures are given without referring to them in the text
· I think, throughout the MS, you have to refer to the skin color of the eggplant fruit, and NOT just mention green eggplant or purple eggplant.
· It has been mentioned that they used the student t-test in Figure 1. Why do you use it to compare multiple means?
· The author did not refer to what has been mentioned in L227-228 in the results section. Then how can you say that it is controlled by a pair of alleles, and then a few lines later, you mention that it is controlled by major and multiple genes? I did not understand this point. Please clarify.
· Please carefully check the Materials and Methods section and provide all necessary information for all your materials used and methodologies followed. Please sequentially provide information, rather than jumping from one topic to another. You will also need to mention all necessary information about different populations' development before moving on to the next subsections, such as DNA extraction, preparation, etc.
· Please draw a concrete conclusion based on your findings.
· Please carefully check your references.
· Additional edits and suggestions are in the manuscript (attached).

Moderate English language edition is required.
Author Response
Thank you for your letter and for the reviewers’ comments concerning our manuscript again. We have studied comments carefully and have made correction which we hope meet with approval. The grammer and word mistake have been corrected in the paper. The reply sentences were bolded and highlight. The main corrections in the paper and the responds to the reviewers’ comments are as flowing:
Reviewer #2
This study deals with the fine mapping of the SmeANS gene, which controls anthocyanin biosynthesis for purple skin in unripe eggplant fruit. The inheritance of eggplant color was studied in F1, F2, and BC1 populations derived from two eggplant lines with different skin colors of the unripe fruit. An SNP in SmANS, EGP22363, was found to be responsible for the green color of unripe eggplant fruit.
The paper addresses an important topic; however, a thorough revision of almost all parts of the MS is needed. I have made my comments and edits directly in the MS file, the most important ones are:
- Please state clearly the objectives of your study, mention in an orderly manner what has been done, and what were the main findings.
Reply: We have made changes in the summary in accordance with your comments.
- I think there are some unnecessary details in the introduction that need to be reduced to focus more on the background and rationale of the study.
Reply: Thank you for your suggestion. We adjusted the introduction to focus more on the background and rationale .
- There are Many mistakes and mismatches in the results.
- Some figures are given without referring to them in the text
- I think, throughout the MS, you have to refer to the skin color of the eggplant fruit, and NOT just mention green eggplant or purple eggplant.
Reply:We have checked the mistakes and mismatches in the results and make corrections. Some figures not mentioned in this article are also illustrated.
- It has been mentioned that they used the student t-test in Figure 1. Why do you use it to compare multiple means?
Reply: I’m sorry for the mistake.The significant differences were determined using ANOVA followed by Tukey’s HSD test (P<0.05). It has been corrected.
- The author did not refer to what has been mentioned in L227-228 in the results section. Then how can you say that it is controlled by a pair of alleles, and then a few lines later, you mention that it is controlled by major and multiple genes? I did not understand this point. Please clarify.
Reply: In this study, we studied the purple and green of eggplant peel without considering the intermediate transition color, and genetic analysis showed that purple and green were dominant traits and controlled by a pair of alleles. But the purple color of eggplant fruit can be divided into purple red, purple black and other different degrees of , different degrees of purple traits are quantitative traits controlled by multiple genes. We added the explanation in discussion.
- Please carefully check the Materials and Methods section and provide all necessary information for all your materials used and methodologies followed. Please sequentially provide information, rather than jumping from one topic to another. You will also need to mention all necessary information about different populations' development before moving on to the next subsections, such as DNA extraction, preparation, etc.
- Please draw a concrete conclusion based on your findings.
- Please carefully check your references.
Reply:The Materials and Methods section had been carefully reviewed and modified.And we carefully summarized the conclusion.
- Additional edits and suggestions are in the manuscript (attached).
We tried our best to improve the manuscript and made some changes in the manuscript. These changes will not influence the content and framework of the paper. We appreciate for Editors/Reviewers’ warm work earnestly, and hope that the correction will meet with approval. Once again, thank you very much for your comments and suggestions.
Round 2
Reviewer 2 Report
Comments and Suggestions for Authors
Dear Authors,
I congratulate you for your great efforts to improve your MS. I think your MS is now much better.
However, I think since this is one of the findings of this study, you need to indicate the way you followed to evaluate the peel color.
In your response, you mentioned that "we studied the purple and green of eggplant peel without considering the intermediate transition color, and genetic analysis showed that purple and green were dominant traits and controlled by a pair of alleles". So, to me, this should be clearly stated when mentioning the mode of inheritance of the peel color.
Please also try to give more informative captions for the figures and Tables.
Some minor edits have been made directly into the MS file (attached).

Minor English language edits are required.
Author Response
Thank you for the reviewers’ comments concerning our manuscript again.
The main corrections in the paper and the responds to the reviewers’ comments are as flowing:
I congratulate you for your great efforts to improve your MS. I think your MS is now much better.
However, I think since this is one of the findings of this study, you need to indicate the way you followed to evaluate the peel color.
Reply: Thank you for your suggestion. In Materials and methods, we indicate the way to evaluate the peel color. Indication are as follows:The plants with a complete absence of purple color of fruit peel in F2 progenies are recorded as ‘green’, otherwise ‘purple’.
In your response, you mentioned that "we studied the purple and green of eggplant peel without considering the intermediate transition color, and genetic analysis showed that purple and green were dominant traits and controlled by a pair of alleles". So, to me, this should be clearly stated when mentioning the mode of inheritance of the peel color.
Reply:We state the reason and method in Result 2.1.
Please also try to give more informative captions for the figures and Tables.
Reply: We have tried to add more detailed informative captions for the figures and Tables.
Some minor edits have been made directly into the MS file (attached).
Reply: The minor edits have been made according to you suggestions.
Once again, thank you very much for your comments and suggestions.
